# MESHINVERSION: 3D TEXTURED MESH RECONSTRUCTION WITH GENERATIVE PRIOR

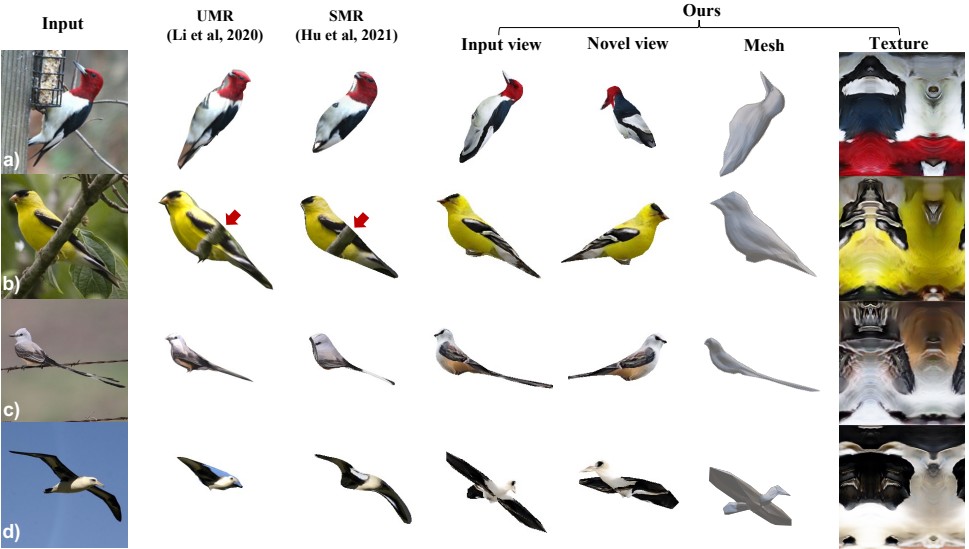

Figure 1: MeshInversion is an alternative approach to single-view textured mesh reconstruction by exploiting generative prior encapsulated in a pre-trained GAN. Our method has three major advantages: **1)** It reconstructs highly faithful and realistic 3D objects, even when observed from novel views; **2)** The reconstruction is robust against occlusion in (b), where red arrows indicate textural errors; **3)** The method generalizes reasonably well to less common shapes, such as birds with (c) extended tails or (d) open wings. Note that both MeshInversion and SMR have access to silhouettes during inference, whereas UMR does not.

## ABSTRACT

Recovering a textured 3D mesh from a single image is highly challenging, particularly for in-the-wild objects that lack 3D ground truths. Prior attempts resort to weak supervision based on 2D silhouette annotations of monocular images. Since the supervision lies in the 2D space while the output is in the 3D space, such in-direct supervision often over-emphasizes the observable part of the 3D textured mesh, at the expense of the overall reconstruction quality. Although previous attempts have adopted various hand-crafted heuristics to reduce this gap, this issue is far from being solved. In this work, we present an alternative framework, **MeshInversion**, that reduces the gap by exploiting the *generative prior* of a 3D GAN pre-trained for 3D textured mesh synthesis. Reconstruction is achieved by searching for a latent space in the 3D GAN that best resembles the target mesh in accordance with the single view observation. Since the pre-trained GAN encapsulates rich 3D semantics in terms of mesh geometry and texture, searching within the GAN manifold thus naturally regularizes the realness and fidelity of the reconstruction. Importantly, such regularization is directly applied in the 3D space, providing crucial guidance of mesh parts that are unobserved in the 2D space. Experiments on standard benchmarks show that our framework obtains faithful 3D reconstructions with consistent geometry and texture across both observed and unobserved parts. Moreover, it generalizes well to meshes that are less commonly seen, such as the extended articulation of deformable objects [1].

---

[1]The code will be available upon paper acceptance.

# 1 INTRODUCTION

Single-view 3D object reconstruction aims at recovering the shape and texture of the object from a single image. This long-standing problem is fundamental to various applications such as robotics navigation, 3D scene understanding, and augmented/virtual reality. A key challenge is the lack of 3D or multi-view supervision due to the prohibitive cost of data collection and annotation for object instances in the wild.

Leveraging more readily available 2D supervisions such as keypoints, Kanazawa et al. (2018b) first introduce CMR, a deep learning framework of textured mesh reconstruction that is trained with a collection of real-world monocular images of an object category. To further relax the supervision constraint, several follow-up studies propose to learn the 3D manifold in a self-supervised manner, only requiring single-view images and their corresponding masks for training (Li et al., 2020; Goel et al., 2020; Bhattad et al., 2021; Hu et al., 2021). This further complicates the problem as minimizing the reconstruction error in the 2D domain often tends to ignore the overall 3D geometry and back-side appearance, which may end up with a shortcut solution that looks plausible only from the input viewpoint. While these methods compensate the relaxed supervision by exploiting various forms of prior information in the CMR-like framework, *e.g.*, semantic invariance in the UV space (Li et al., 2020) and interpolated consistency of the predicted 3D attributes (Hu et al., 2021), this task remains challenging and worth further exploration.

In this work, we propose an alternative approach that is built upon generative prior possessed by Generative Adversarial Networks (GANs), named **MeshInversion**. GANs are typically known for their exceptional ability to capture comprehensive knowledge (Brock et al., 2019; Karras et al., 2019; Shu et al., 2019), empowering the success of *GAN inversion* in various image restoration tasks (Gu et al., 2020; Pan et al., 2020b) and in point cloud completion (Zhang et al., 2021). By training a GAN to synthesize 3D shapes in the form of a topology-aligned texture and deformation map, one could enable the generator to capture rich prior knowledge of a certain object category, including high-level semantics, object geometries, and texture details. We, therefore, propose to exploit this appealing generative prior through GAN inversion. Formally, our framework finds the latent code of the GAN manifold that best recovers the 3D object corresponding to the input image with a pre-trained 3D GAN. Given the single-view observation, the latent code is updated towards minimizing 2D reconstruction losses by rendering the 3D object onto the image plane. Hence, the 3D GAN manifold *implicitly* constrains the reconstructed 3D shape within the realistic boundaries, whereas minimization of the existing 2D losses *explicitly* drives the 3D shape towards a faithful reflection of the input image.

Searching for the optimal latent code in the GAN manifold for single-view 3D object reconstruction is not trivial. Specifically, reprojection misalignment easily leads to a local minimum, and the impact of discretization-induced information loss is amplified by the smooth manifold. To address the misalignment issue, we propose a **Chamfer Texture Loss**, which relaxes the one-to-one pixel correspondences in existing low-level losses and allows the match to be found within a local region. By jointly considering the appearance and positions of image pixels or feature vectors, it provides a robust appearance distance despite inaccurate camera poses and in the presence of high-frequency textures. To avoid information loss, we propose a **Chamfer Mask Loss**, which intercepts the rasterization process and computes the Chamfer distance between the projected vertices before discretization to retain information, with the foreground pixels of the input image projected to the same continuous space. Hence, it becomes more sensitive to small variations in shape and offers a more accurate gradient for geometric learning.

Our proposed method demonstrates compelling performance for 3D reconstruction from real-world monocular images. Overall, MeshInversion gives highly plausible and faithful 3D reconstruction in terms of both appearance and 3D shape. It achieves state-of-the-art results on the perceptual metric, *i.e.*, FID, when evaluating the textured mesh from various viewpoints, and is on-par with the existing CMR-based methods in terms of geometric accuracy. In addition, MeshInversion benefits from a holistic understanding of the objects given the generative prior. As a result, it not only gives a realistic recovery of the back-side texture but also generalizes well in the presence of occlusion. Furthermore, MeshInversion also shows fairly remarkable generalization for 3D shapes that are less commonly seen, such as birds with open wings and long tails. Some representative reconstruction results are depicted in Fig. 1.

## 2 RELATED WORK

**Single-view 3D Reconstruction.** It recovers the 3D information of an object, such as its shape and texture, from a single-view observation. Several methods use image-3D object pairs (Wang et al., 2018; Pan et al., 2019; Mescheder et al., 2019; Rematas et al., 2021) or multi-view images (Niemeyer et al., 2020; Liu et al., 2019; Yariv et al., 2020; Wang et al., 2021; Oechsle et al., 2021) for training, which limit the scenarios to synthetic data. Another line of work fits the parameters of a 3D prior morphable model, *e.g.*, SMPL for humans and 3DMM for faces (Gecer et al., 2019; Sanyal et al., 2019; Kanazawa et al., 2018a), which are expensive to build and difficult to extend to many different natural object categories. To relax the constraints on supervision, CMR (Kanazawa et al., 2018b) reconstructs category-specific textured mesh by training with a collection of monocular images and associated 2D supervisions, *i.e.*, 2D key-points, camera poses, and silhouette masks. Thereafter, several follow-up works further relax the supervision, *e.g.*, masks only, and improves the reconstruction results mainly by exploiting forms of prior information across the object category or specific to the CMR-like framework. Specifically, they incorporate the prior by enforcing various types of cycle consistencies, such as texture cycle consistency (Li et al., 2020; Bhattad et al., 2021), rotation adversarial cycle consistency (Bhattad et al., 2021), and interpolated consistency (Hu et al., 2021). Some of these methods also leverage external information, *e.g.*, category-level mesh templates (Goel et al., 2020; Bhattad et al., 2021), and semantic parts provided by an external SCOPS model (Li et al., 2020).

For texture modeling, a direct regression of pixel values in the UV texture map often leads to blurry images, *e.g.*, Goel et al. (2020). Therefore, the mainstream approach is to regress pixel coordinates, *i.e.*, learning *texture flow* from the input image to the texture map. Although texture flow is easier to regress and usually provides a vivid front view result, it is often unable to capture the high-level semantics, and thus fails to generalize well to novel views or occluded regions. Our approach directly predicts the texture pixel values by incorporating a pre-trained GAN. In contrast to the texture flow approach, it benefits from a holistic understanding of the objects given the generative prior and offers high plausibility and fidelity at the same time.

**GAN Inversion.** A well-trained GAN usually captures useful statistics and semantics underlying the training data. In the 2D domain, GAN prior has been explored extensively in various image restoration and editing tasks (Bau et al., 2019b; Pan et al., 2020b; Gu et al., 2020). GAN inversion, the common method in this line of work, finds a latent code that best reconstructs the given image using the pre-trained generator. Typically, the target latent code can be obtained via gradient descent (Ma et al., 2018; Lipton & Tripathi, 2017), projected by an additive encoder that learns the inverse mapping of a GAN (Bau et al., 2019a), or a combination of them (Zhu et al., 2020). Recently, there are attempts to apply GAN inversion in the 3D domain. Zhang et al. (2021) use a pre-trained point cloud GAN to address shape completion in the canonical pose, giving remarkable generalization for out-of-domain data such as real-world partial scans. Pan et al. (2020a) first explore to recover the geometric cues from pre-trained 2D GANs and achieve exceptional reconstruction results, but the reconstructed shapes are limited to 2.5D due to limited poses that 2D GANs can synthesize. In this work, we directly exploit the prior from a 3D GAN to reconstruct the shape and texture of complete 3D objects.

## 3 APPROACH

**Preliminaries.** We represent a 3D object as a textured triangle mesh $\mathbf{O} \equiv (\mathbf{V}, \mathbf{F}, \mathbf{T})$, where $\mathbf{V} \in \mathbb{R}^{|\mathbf{v}| \times 3}$ represents the location of the vertices, $\mathbf{F}$ represents the faces that define the fixed connectivity of vertices in the mesh, and $\mathbf{T}$ represents the texture map. An individual mesh is isomorphic to a 2-pole sphere, and thus we model the deformation $\Delta \mathbf{V}$ from the initial sphere template, and then obtain the final vertex positions by $\mathbf{V} = \mathbf{V}_{sphere} + \Delta \mathbf{V}$. Previous methods (Kanazawa et al., 2018b; Li et al., 2020; Goel et al., 2020) typically regress individual vertices via a fully connected network (MLP). In contrast, recent studies have found that using a 2D convolutional neural network (CNN) to learn a deformation map in the UV space would benefit from consistent semantics across the entire category (Pavllo et al., 2020; Bhattad et al., 2021). In addition, the deformation map $\mathbf{S}$ and the texture map $\mathbf{T}$ are topologically aligned, so both the values can be mapped to the mesh via the same UV mapping. We assume a weak-perspective camera projection, where the camera pose $\mathbf{c}$ is parameterized by scale $\mathbf{s} \in \mathbb{R}$, translation $\mathbf{t} \in \mathbb{R}^2$, and rotation in the form of quaternion $\mathbf{r} \in \mathbb{R}^4$. We use the DIB-R by Chen et al. (2019) as our differentiable renderer. We denote $\mathbf{I} = R(\mathbf{S}, \mathbf{T}, \mathbf{c})$ as the image rendering process.

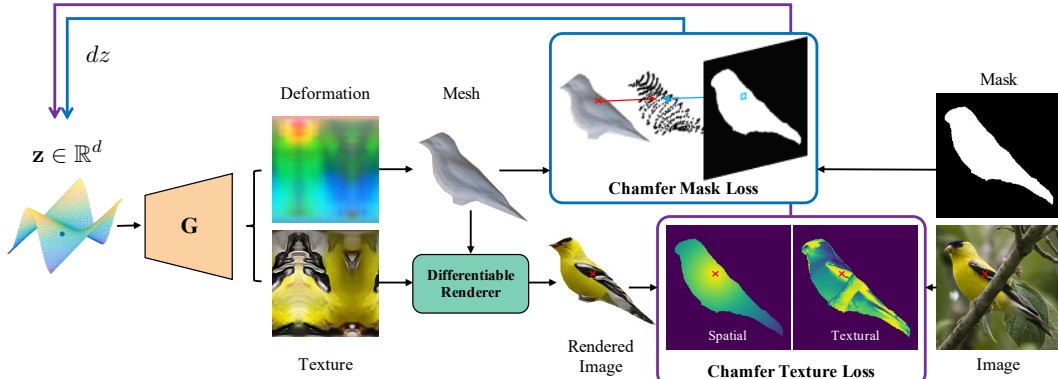

Figure 2: **MeshInversion framework.** We exploit a pre-trained 3D GAN that synthesizes plausible textured meshes in the form of topology-aligned deformation map and texture map in the UV space. Given a single-view image and the mask, we search for the latent code that minimizes reconstruction losses in the 2D domain via gradient descent. We address the intrinsic challenges associated with rendering with two Chamfer-based losses: 1) **Chamfer Texture Loss** (Sec 3.2) relaxes the one-to-one pixel correspondences between two images or their respective feature maps and allows the match to be found within a local region, in which a pairwise distance is factorized into a spatial distance term and a textural distance term. We illustrate the distance maps between one anchor point from the rendered image to the target image, where brighter regions correspond to smaller distances. 2) **Chamfer Mask Loss** (Sec 3.3) intercepts the discretization process and computes the chamfer distance between the projected vertices and the foreground pixels projected to the same continuous space.

## 3.1 MESHINVERSION

In this work, we propose to exploit generative prior in the task of single-view 3D reconstruction. Pavllo et al. (2020) recently propose a CNN-based GAN framework, ConvMesh, which generates 3D objects in the form of topology-aligned texture maps and deformation maps in the UV space. Despite being trained with purely 2D supervisions from single-view natural images, ConvMesh captures meaningful prior knowledge of the object category (such as high-level semantics, object geometries, and texture details), and is able to generate objects with a wide range of plausible shapes and high-fidelity textures. Motivated by the success of GAN inversion in various image restoration tasks, we incorporate the prior stored in a pre-trained ConvMesh to achieve plausible and faithful recovery of 3D shape and appearance through GAN inversion.

**Pre-training Stage.** Prior to GAN inversion, we first pre-train the textured GAN on the training split to capture desirable prior knowledge for 3D reconstruction. Pavllo et al. (2020) train ConvMesh in the UV space, where pseudo ground truths of texture map and deformation map are obtained by inverse rendering from the single-view images. In addition to the UV space discrimination, we further enhance the photorealism of the generated 3D objects by introducing a discriminator in the image space, following the architecture of PatchGAN as in Isola et al. (2017). The loss functions for the pre-training stage are shown as follows. $D_{uv}$ refers to the discriminator in the UV space; $D_I$ refers to the discriminator in the image space; $\lambda_{uv}$ and $\lambda_I$ are the weights. We use least-squares losses following (Mao et al., 2017). Ablation study on image space discrimination, and more details on ConvMesh and pseudo ground truths preparation can be found in the Supplementary Material.

$$\mathcal{L}_G = \lambda_{uv}\mathbb{E}_{\mathbf{z}\sim P_{\mathbf{z}}}[(D_{uv}(G(\mathbf{z}))-1)^2] + \lambda_I\mathbb{E}_{\mathbf{z}\sim P_{\mathbf{z}}(\mathbf{z})}[(D_I(R(G(\mathbf{z})),\mathbf{c})-1)^2] \quad (1)$$

$$\mathcal{L}_{D_{uv}} = \mathbb{E}_{\mathbf{S},\mathbf{T}\sim P_{pseudo}}[(D_{uv}(\mathbf{S},\mathbf{T})-1)^2] + \mathbb{E}_{\mathbf{z}\sim P_{\mathbf{z}}(\mathbf{z})}[(D_{uv}(G(\mathbf{z})))^2] \quad (2)$$

$$\mathcal{L}_{D_I} = \mathbb{E}_{\mathbf{I}\sim P_{data}}[(D_I(\mathbf{I})-1)^2] + \mathbb{E}_{\mathbf{z}\sim P_{\mathbf{z}}(\mathbf{z})}[(D_I(R(G(\mathbf{z})),\mathbf{c})^2] \quad (3)$$

**Inversion Stage.** We now formally introduce GAN inversion for single-view 3D reconstruction. Given a pre-trained ConvMesh that generates a textured mesh from a latent code, $\mathbf{S}, \mathbf{T} = G(\mathbf{z})$, we aim to find the $\mathbf{z}$ that best recovers the 3D object from the input image $\mathbf{I}_{in}$ and its silhouette mask $\mathbf{M}_{in}$. Specifically, we achieve this by test-time optimization, where $\mathbf{z}$ is optimized towards minimizing the overall reconstruction loss $\mathcal{L}_{inv}$ via gradient descent. In general, the inversion stage can be denoted by Eq. 4 and illustrated in Fig. 2.

$$\mathbf{z}^* = \arg\min_{\mathbf{z}} \mathcal{L}_{inv}(R(G(\mathbf{z}),\mathbf{c}),\mathbf{I}_{in}) \quad (4)$$

Given the single-view image and the associated mask, we would need to project the reconstructed 3D object to the observation space for computing $\mathcal{L}_{inv}$. However, such 3D-to-2D degradation is non-trivial. In existing image-based tasks involving GAN inversion, we always assume pixel-wise image correspondence in the observation space. This is because both the appearance and pose of the scene can be controlled in the latent space of 2D GANs (Shen et al., 2020; Shen & Zhou, 2021). In contrast, we generate 3D objects in the form of a pose-invariant UV texture map and deformation map, and explicitly control the object pose during differentiable rendering. While concurrently optimizing the latent code and the camera pose is a plausible approach, this often suffers from camera-shape ambiguity (Li et al., 2020) and leads to erroneous reconstruction. In addition, the presence of high-frequency textures, *e.g.*, complex bird feathers, often leads to blurry appearance even with slight discrepancies in pose. Consequently, it is infeasible to assume a perfect alignment between the rendered image and the input one, which calls for a robust form of appearance loss in Sec 3.2.

## 3.2    CHAMFER TEXTURE LOSS

To facilitate a search in the GAN manifold without worrying about blurry reconstructions, we reconsider the appearance loss by relaxing the pixel-aligned assumption in existing low-level losses. Taking inspiration from the point cloud data structure, we treat a 2D image as a set of 2D colored points, which have both appearance attributes, *i.e.*, RGB values, and spatial attributes, the values of which relate to their coordinates in the image grid. Thereafter, we aim to measure the dissimilarity between the two colored point sets via Chamfer distance,

$$\mathcal{L}_{CD}(\mathbb{S}_1, \mathbb{S}_2) = \frac{1}{|\mathbb{S}_1|} \sum_{x \in \mathbb{S}_1} \min_{y \in \mathbb{S}_2} \mathbf{D}_{xy} + \frac{1}{|\mathbb{S}_2|} \sum_{y \in \mathbb{S}_2} \min_{x \in \mathbb{S}_1} \mathbf{D}_{yx} \tag{5}$$

Intuitively, defining the pairwise distance between pixel $x$ and pixel $y$ in the two respective images should jointly consider their appearance and location. In this regard, we factorize the overall pairwise distance $\mathbf{D}_{xy}$ into an appearance term $\mathbf{D}_{xy}^a$ and a spatial term $\mathbf{D}_{xy}^s$, both of which are L2 distance. Like conventional Chamfer distance, single-sided pixel correspondences are determined by column-wise or row-wise minimum in the distance matrix $\mathbf{D}$.

Importantly, we desire the loss to be tolerant and only tolerant of local misalignment, as large misalignment will potentially introduce noisy pixel correspondences that may jeopardize appearance learning. Inspired by the focal loss for detection (Lin et al., 2017), we introduce an exponential operation in the spatial term to penalize those spatially distant pixel pairs. Therefore, we define the overall distance matrix $\mathbf{D}$ as follows

$$\mathbf{D} = \max((\mathbf{D}^s + \epsilon_s)^\alpha, 1) \times (\mathbf{D}^a + \epsilon_a) \tag{6}$$

where $\mathbf{D}^a$ and $\mathbf{D}^s$ are the appearance distance matrix and spatial distance matrix respectively; $\epsilon_s$ and $\epsilon_a$ are residual terms to avoid incorrect matches with identical location or identical pixel value respectively; $\alpha$ is the scaling factor for flexibility. Specifically, we let $\epsilon_s < 1$ so that the spatial term remains one when two pixels are slightly misaligned. Note that the spatial term is not differentiable and it only serves as a weight matrix for appearance learning.

By substituting the resulting $\mathbf{D}$ into Eq. 5, we thus have the final formulation of our proposed **Chamfer Texture Loss**, denoted as $\mathcal{L}_{CT}$. Such relaxed formulation provides a robust measure of texture distance, which effectively eases searching of the target latent code while preventing blurry reconstructions; in return, although $\mathcal{L}_{CT}$ only concerns about local patch statistics but not photorealism, the use of GAN prior is sufficient to give realistic predictions. Besides, the GAN prior also allows computing $\mathcal{L}_{CT}$ with a down-sampled size of colored points. In practice, we randomly select 8096 pixels from each image as a point set.

Moreover, the proposed formulation gives flexible control between appearance and spatial attributes, which is readily extendable to misaligned feature maps to achieve more semantically faithful 3D reconstruction. Specifically, we apply the Chamfer texture loss between the (foreground) feature maps extracted with a pre-trained VGG-19 network (Simonyan & Zisserman, 2014) from the rendered image and the input image. It is worth noting that the feature-level Chamfer texture loss  is somewhat related to the contextual loss (Mechrez et al., 2018), which addresses the misalignment issue for image transfer. The key difference is that the contextual loss only considers the feature distances but ignores their locations. We compare against the contextual loss in the experiment.

|  | CMR | U-CMR | UMR | SMR | View-gen[†] | Ours |
|---|---|---|---|---|---|---|
| IoU ↑ | 0.703 | 0.701 | 0.734 | **0.8** | 0.629 | 0.708 |
| $\text{FID}_1$ ↓ | 140.9 | 65.0 | 40.0 | 52.9 | - | **38.6** |
| $\text{FID}_{10}$ ↓ | 176.2 | 314.9 | 72.8 | 63.2 | - | **38.6** |
| $\text{FID}_{12}$ ↓ | 180.1 | 315.2 | 86.9 | 79.2 | 70.3 | **56.6** |

Table 1: Quantitative results on CUB. [†]: We report results from Bhattad et al. (2021) as the implementation is not available.

## 3.3 CHAMFER MASK LOSS

Conventionally, the geometric distance is usually computed between two binary masks in terms of L1 or IoU loss (Kanazawa et al., 2018b; Goel et al., 2020; Li et al., 2020; Hu et al., 2021). However, obtaining the mask of the reconstructed mesh usually involves rasterization that discretizes the mesh into a grid of pixels. This operation inevitably introduces information loss and thus inaccurate supervision signals. This is particularly harmful to a well-trained ConvMesh, the shape manifold of which is typically smooth. Specifically, a small perturbation in $\mathbf{z}$ usually corresponds to a slight variation in the 3D shape, which may translate to an unchanged binary mask. This usually leads to an insensitive gradient for back-propagation, which undermines geometric learning.

To this end, we propose a **Chamfer Mask Loss**, or $\mathcal{L}_{CM}$, to compute the geometric distance in an unquantized 2D space. Instead of rendering the mesh into a binary mask, we directly project the 3D vertices of the mesh onto the image plane, $\mathbb{S}_v = P(\mathbf{S}, \mathbf{T}, \mathbf{c})$. For the foreground mask, we obtain the positions of the foreground pixels by normalizing their pixel coordinates in the range of $[-1, 1]$. Thereafter, we compute the Chamfer distance between $\mathbb{S}_v$ and $\mathbb{S}_f$, which is the set of normalized coordinates of the input mask. Note that the bidirectional Chamfer distance between the sparse set $\mathbb{S}_v$ and dense set $\mathbb{S}_f$ would regularize the vertices from highly uneven deformation.

## 3.4 OVERALL OBJECTIVE FUNCTION

We apply the pixel-level Chamfer texture loss $\mathcal{L}_{CT-p}$ and the feature-level one $\mathcal{L}_{CT-f}$ as our appearance losses, and the Chamfer mask loss $\mathcal{L}_{CM}$ as our geometric loss. Besides, we also introduce two regularizers: the smooth loss $\mathcal{L}_{smooth}$ that encourages neighboring faces to have similar normals, *i.e.*, low cosine; the latent space loss $\mathcal{L}_z$ that regularizes the L2 norm of $\mathbf{z}$ to ensure Gaussian distribution. The overall object function is shown in Eq. 7, where $\lambda_{CT-p}, \lambda_{CT-f}, \lambda_{CM}, \lambda_{smooth}$, and $\lambda_z$ are the weights, $g$ is the pre-trained VGG-19 feature extractor, and $\odot$ denotes element-wise product.

$$\mathcal{L}_{inv} = \lambda_{CT-p}\mathcal{L}_{CT-p}(R(\mathbf{S}, \mathbf{T}, \mathbf{c}), \mathbf{I}_{in} \odot \mathbf{M}_{in}) + \lambda_{CT-f}\mathcal{L}_{CT-f}(g(R(\mathbf{S}, \mathbf{T}, \mathbf{c})), g(\mathbf{I}_{in} \odot \mathbf{M}_{in}))$$
$$+ \lambda_{CM}\mathcal{L}_{CM}(P(\mathbf{S}, \mathbf{T}, \mathbf{c}), \mathbf{M}_{in}) + \lambda_{smooth}\mathcal{L}_{smooth}((\mathbf{V}, \mathbf{F})) + \lambda_z\mathcal{L}_z(\mathbf{z})$$
$$(7)$$

## 4 EXPERIMENTS

**Datasets and Experimental Setting.** We primarily evaluate our method on CUB-200-2011 dataset (Wah et al., 2011). It consists of 200 species of birds with a wide range of shapes and feathers, making it an ideal benchmark to evaluate 3D reconstruction in terms of both geometric and texture fidelity. Apart from the organic shapes like birds, we also validate our method on 11 man-made rigid car categories from PASCAL3D+ (Xiang et al., 2014).

We use the same train-validation-test split as provided by CMR (Kanazawa et al., 2018b). The images in both datasets are annotated with foreground masks and camera poses. Specifically, we pre-train ConvMesh on the pseudo ground truths derived from the training split following a class conditional setting (Pavllo et al., 2020). During inference, MeshInversion has access to the input image and the associated mask and camera pose. Since the focus of this work is to study the effectiveness of exploiting GAN prior in 3D reconstruction, we base our experiments mainly on silhouette ground truths and cameras *estimated* by structure-from-motion (SfM), following the setting of SMR (Hu et al., 2021) and Chen et al. (2019) respectively. In the Supplementary Material, we also validate that our method is reasonably robust to inaccurate camera poses predicted by an off-the-shelf camera pose estimator from Kanazawa et al. (2018b), and is invariant under masks predicted by instance segmentation method PointRend (Kirillov et al., 2020) pre-trained on COCO (Lin et al., 2014).

**Evaluation Strategy.** Since there are no 3D ground truths available, we evaluate our method against various baselines from the following three aspects: 1) We evaluate the geometry accuracy in the 2D domain by IoU between the input mask and the rendered mask. 2) We evaluate the appearance quality by image synthesis metric FID (Heusel et al., 2017), which compares the distribution of test set images and the render of reconstruction. We report both single-view FID ($FID_1$) and multi-view FIDs. Specifically, we report $FID_{12}$ following the setting in SMR, which covers azimuth from $0°$ to $360°$ at an interval of $30°$. However, we notice that the exact front view ($90°$) and the exact back view ($270°$) are rarely seen in CUB, so we mainly report $FID_{10}$ that excludes these two viewpoints. 3) Apart from extensive qualitative results, we conduct a user study to evaluate human preferences comparing our method against existing baselines, in terms of both shape and appearance.

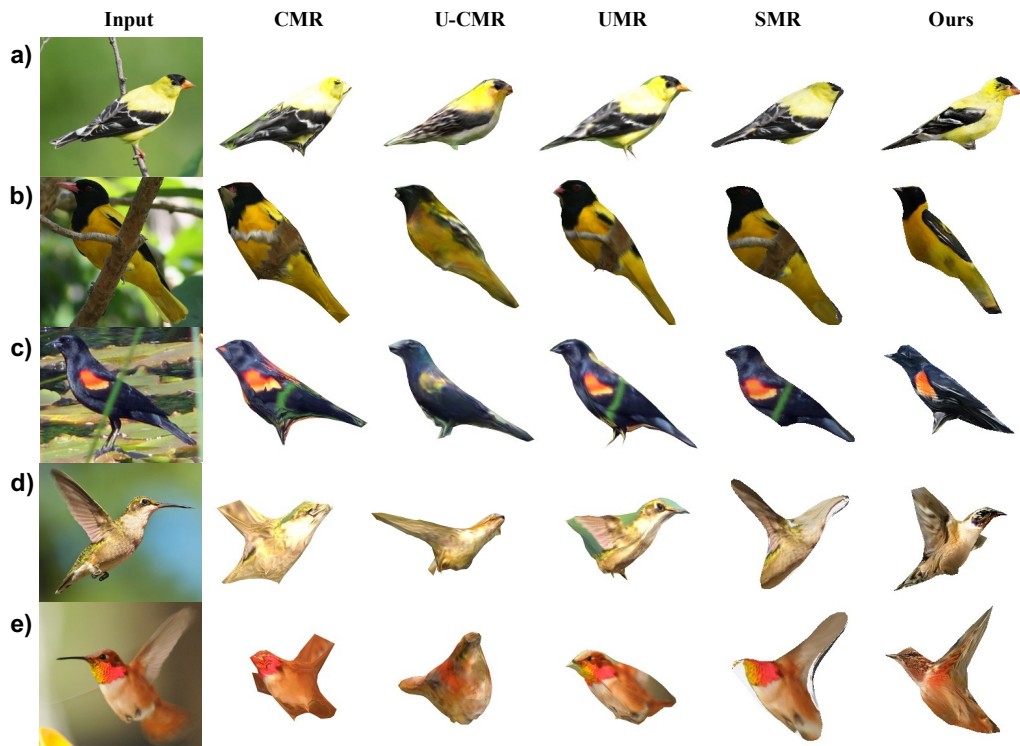

Figure 3: Qualitative results on CUB. MeshInversion achieves highly faithful and realistic 3D reconstruction. In particular, it exhibits superior generalization under various challenging scenarios, including with occlusion (b, c) and extended articulation (d, e).

## 4.1 COMPARISON WITH BASELINES

We compare MeshInversion with various CMR-based methods on the CUB dataset. Among these methods, only U-CMR (Goel et al., 2020) directly predicts texture maps. CMR (Kanazawa et al., 2018b), UMR (Li et al., 2020), and SMR (Hu et al., 2021) all use texture flow. We provide quantitative results in Tab. 1. Overall, our method achieves state-of-the-art results on perceptual metrics, particularly multi-view FIDs, and is on par with existing baselines in terms of IoU.

We provide single-view and novel-view qualitative results in Fig. 3 and Fig. 4, respectively. It is observed that our method achieves highly faithful and realistic 3D reconstruction, particularly when observed from novel views. In contrast, although texture flow-based methods generally give superior texture reconstruction for visible regions, they tend to give incorrect predictions for invisible regions, *e.g.*, abdomen or back. Moreover, our method benefits from a holistic understanding of the objects and gives remarkable performance in the presence of occlusion, while texture flow-based methods only learn to copy from the foreground pixels including the obstacles, *e.g.*, twig, from the bird. Due to the same reason, these methods also tend to copy background pixels onto the reconstructed object when the shape prediction is inaccurate. Moreover, our method generalizes reasonably well to highly articulated shapes, such as birds with long tails and open wings, where many of the existing methods fail to give satisfactory reconstructions. See the Supplementary Material for more novel-view results.

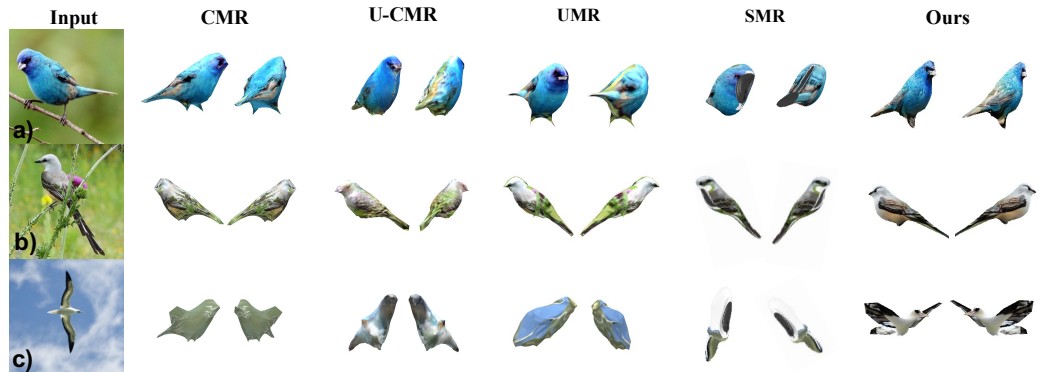

Figure 4: Novel-view rendering results on CUB for MeshInversion and various baselines. In general, our method gives plausible 3D reconstruction even when observed from novel views, whereas many of these baseline methods fail to generalize well to invisible regions and challenging articulations.

| Criterion | CMR | U-CMR | UMR | SMR | Ours |
|---|---|---|---|---|---|
| Texture | 2.7% | 15.7% | 13.2% | 6.6% | **61.8%** |
| Shape | 2.7% | 19.6% | 14.6% | 4.2% | **58.9%** |
| Overall | 2.5% | 19.8% | 12.8% | 3.3% | **61.5%** |

Table 2: User preference study on CUB in terms of the quality and faithfulness of texture, shape, and overall 3D reconstruction.

To further evaluate the 3D reconstruction results, we conduct a user preference study on 30 selected images from the test set. The reconstructed 3D objects by different methods are rendered from three different viewpoints, for the users to choose the most realistic and faithful reconstruction in terms of texture, shape, and overall 3D reconstruction. The evaluation results based on 40 users' responses in Tab. 2 show that MeshInversion achieves the most preferred 3D reconstruction, whereas all texture flow-based methods give poor results mainly due to their incorrect prediction for unseen regions.

However, it is worth noting that all the baseline methods are regression-based, and all except for SMR have no access to silhouettes at test time. For a fair comparison, we also conduct test-time optimization for existing baselines with access to silhouettes during inference in the Supplementary Material, and the results show that our method remains highly competitive especially for FIDs.

## 4.2 TEXTURE TRANSFER

As the shape and texture are topologically and semantically aligned in the UV space, it allows us to modify the surface appearance across bird instances. As shown in Fig. 5, we sample pairs of instances and transfer the texture between one another, by swapping their texture maps. Note that even for extended articulations like open wings and long tails, the resulting new 3D objects remain highly faithful and realistic.

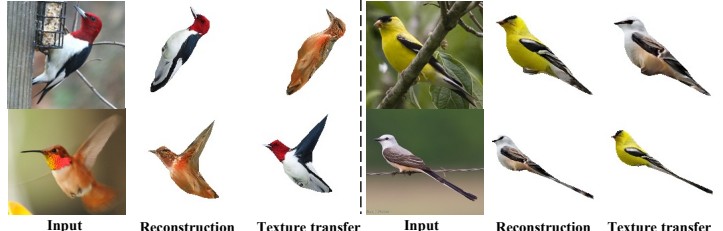

Input     Reconstruction     Texture transfer     Input     Reconstruction     Texture transfer

Figure 5: MeshInversion enables faithful and realistic texture transfer between bird instances even with highly articulated shapes.

## 4.3 EVALUATION ON PASCAL3D+ CAR

We also evaluate our method on the man-made rigid car category. As demonstrated in Fig. 6, our method performs reasonably well across different car models and appearances. Unlike Bhattad et al. (2021) and Goel et al. (2020) that explicitly use one or multiple mesh templates provided by PASCAL3D+, the appealing GAN prior implicitly provides a rich number of templates that

| Mask loss | Texture loss | IoU ↑ | $FID_1$ ↓ | $FID_{10}$ ↓ |
|---|---|---|---|---|
| | L1 loss | 0.705 | 71.2 | 97.3 |
| | L2 loss (MSE) | **0.708** | 75.2 | 108.1 |
| $L_{CM}$ | perceptual loss | 0.701 | 49.8 | 52.3 |
| | contextual loss | 0.699 | 65.2 | 72.5 |
| | L1 + percep loss | 0.707 | 47.1 | 50.8 |
| L1 loss | $L_{CT-p} + L_{CT-f}$ | 0.582 | 51.7 | 53.2 |
| IoU loss | $L_{CT-p} + L_{CT-f}$ | 0.605 | 51.2 | 50.8 |
| $L_{CM}$ | $L_{CT-p} + L_{CT-f}$ | **0.708** | **38.6** | **38.6** |

Table 3: Ablation study. Our proposed Chamfer texture losses $\mathcal{L}_{CT-p}$ and $\mathcal{L}_{CT-f}$, and Chamfer mask loss $\mathcal{L}_{CM}$ are effective to address the misalignment and quantization challenges induced by rendering.

makes it possible to reconstruct cars of various models. More novel-view results can be found in the Supplementary Material.

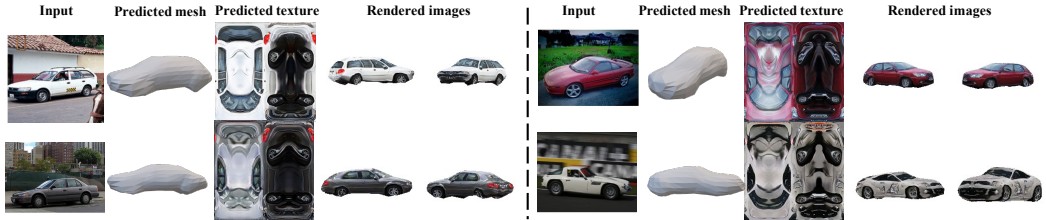

Figure 6: Qualitative results on PASCAL3D+ Car. Our method gives reasonably good performance across different car models and appearances.

## 4.4 ABLATION STUDY

We evaluate the effectiveness of the proposed losses in our MeshInversion framework by comparing with conventional texture and mask losses, as reported in Tab. 3. We compare our Chamfer texture loss, $\mathcal{L}_{CT}$, with pixel-to-pixel L1 and L2 losses, perceptual loss (Johnson et al., 2016), and contextual loss (Mechrez et al., 2018). Both L1 and L2 losses tend to give blurry reconstructions, as perfect image pixel-wise alignment is not guaranteed even with the not-so-accurate ground truth cameras *estimated* by structure-from-motion. Feature-based losses are generally more robust to misalignment, but they are usually not discriminative enough to reflect the appearance details between the two images. In particular, although the contextual loss is designed to address the misalignment issue in image-to-image translation tasks, it only considers feature distances while ignoring their positions in the image. In contrast, our proposed pixel-level and feature-level Chamfer texture losses achieve highly faithful and realistic texture reconstruction, and gives the best single-view and multi-view FID scores. We compare the Chamfer mask loss, $\mathcal{L}_{CM}$, with L1 mask loss and IoU loss (negative intersection over union loss). $\mathcal{L}_{CM}$ avoids quantization during 3D-to-2D degradation, resulting in more accurate gradients for geometric learning. Consequently, it yields the highest IoU result. More ablation studies can be found in the Supplementary Material.

## 5 DISCUSSION

We have presented MeshInversion for single-view 3D object reconstruction. It exploits generative prior encapsulated in a pre-trained GAN and reconstructs textured shapes through GAN inversion. To address reprojection misalignment and discretization-induced information loss due to 3D-to-2D degradation, we propose two Chamfer-based losses in the 2D space, *i.e.*, Chamfer texture loss and Chamfer mask loss. By efficiently incorporating the GAN prior, MeshInversion achieves highly realistic and faithful 3D reconstruction, and exhibits superior generalization power for challenging cases, such as in the presence of occlusion or extended articulations. However, this challenging problem is far from being solved. In particular, although we can faithfully reconstruct flying birds with open wings, the wings are only represented by a few vertices due to semantic consistency across the entire category, which strictly limits the representation power in terms of geometry and texture details. Therefore, future work may explore more flexible solutions, for instance, an adaptive number of vertices can be assigned to articulated regions to accommodate richer details.

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
