# OpenReview forum: "MeshInversion: 3D textured mesh reconstruction with generative prior"
_ICLR.cc/2022/Conference — ICLR 2022 Submitted_

### Official Review · Reviewer_i1Mn · 2021-10-25

**Correctness:** 2
**Technical Novelty And Significance:** 3
**Empirical Novelty And Significance:** 3
**Recommendation:** 5
**Confidence:** 3

**Details Of Ethics Concerns:**

N.A.

**Main Review:**

Strength
- The idea of optimizing the 3D shape via a pre-trained 3D GAN is interesting and sensible. It effectively leverages 3D priors for the task of single-view reconstruction of 3D shapes. Prior work mostly use explicit regularizations, such as as-rigid-as-possible or L1 losses to avoid irregular deformation of the shapes. In contrast, MeshInversion regularizes the deformation in a implicit but meaningful way -- the deformed shape has to lie on the 3D GAN manifold.
- The writing is mostly clear and is easy to follow.

Weakness
- One major weakness is the evaluation protocol. As described, the proposed method takes a test RGB and silhouette image as input to optimize the 3D shape. However, the same inputs are used as the "ground-truth" to evaluate the optimized 3D shapes (Table 2). This does not seem to be a good evaluation metric, considering a method could overfit to the ground-truth masks and RGB image without reconstructing a meaningful 3D shape. A standard way to quantitatively evaluate the method would be using ground-truth 3D data as did in CMR. Another way is to evaluate keypoint transfer accuracy as in CSM [1].
- Another weakness is the unfair comparison. For instance in Table 2, the baseline methods (such as CMR, UCMR) do not take test time image and object silhouette as inputs, but MeshInversion does. A fair comparison would be also test-time optimize CMR, UCMR, etc. Considering MeshInversion has access to test silhouette image but the baselines do not, Figure 1 caption may also needs to reflect this.
- I'm also not convinced that the proposed chamfer loss is more powerful than the widely adopted differentiable rendering-based reconstruction losses. As in Sec 3.3, how would differentiable rasterization "introduce information loss"? Moreover, how would the proposed chamfer loss deal with occlusion, which should not be a problem for differentiable rendering? For instance, the proposed chamfer texture loss would project both the front and back surface points to the image, and compare with the observed pixels. Would it color the back surface with observed pixels and is this desired?

Minor comments / questions
- When optimizing the latent shape/texture code on a single image, are the weights of the generative network frozen? How much does the accuracy of the pre-trained network affect the test-time optimization? The extreme case is using a randomly initialized network for optimization.
- The ablation result (Table 1) should normally go after the results (Table 2)

[1] Kulkarni, Nilesh, Abhinav Gupta, and Shubham Tulsiani. "Canonical surface mapping via geometric cycle consistency." Proceedings of the IEEE/CVF International Conference on Computer Vision. 2019.

**Summary Of The Paper:**

The paper proposes MeshInversion, a method for single-view 3D shape reconstruction. Given a test image with silhouette, it optimizes the 3D shape represented as the input latent code to a pre-trained 3D GAN (specifically ConvMesh, NeurIPS 20) via the proposed chamfer losses. The idea of leveraging the 3D GAN shape priors to regularize the reconstruction of unobservable surface is novel and sensible. The method is quantitatively evaluated on CUB and performs better than some recent methods on FID scores and user study.

**Summary Of The Review:**

The paper proposes an interesting idea for single-view 3D shape reconstruction using 3D shape priors built-in a GAN model. However, the experiments and comparisons are not convincing enough to support the major claims. A few technical details also needs to be clarified.

---

> ### Author Response · Authors · 2021-11-23
> **Reply to Reviewer i1Mn [1/2]**
>
> Thank you for your constructive comments. Below we address individual concerns. We also include our revision in the main paper and the supplementary material(content in purple).
>
> **Q1: One major weakness is the evaluation protocol. RGB and silhouette images are both the inputs during inference and the "ground-truth" for evaluation. The use of ground-truth 3D data and keypoint transfer accuracy as in CSM for evaluation.**
>
> A1: Thanks for the alternative evaluation metrics. While ground-truth 3D data is ideal for evaluation, they are not available in the CUB dataset. Keypoint transfer accuracy proposed in CSM can be used for texture flow-based methods, such as CMR and UMR, as the learned texture flow can also map the keypoint locations from the input image to the reconstructed mesh and then reproject to the rendered image, for computing keypoint transfer accuracy during evaluation. In contrast, our method directly predicts the RGB value in the UV texture map, and it does not involve keypoints. Therefore, keypoint transfer accuracy is not applicable to evaluate our method.
>
> To further assess if our method and others reconstruct meaningful and faithful 3D shapes, we enhance our evaluation in the following aspects in Section 4.1: 1) We quantitatively evaluate the multi-view FID to make sure if the rendered images look realistic from different viewpoints. 2) We provide qualitative results from more novel views, with more results in Section 7 of the revised Supplementary Material. 3) We conduct a user study to evaluate the quality and faithfulness of texture, shape, and overall 3D reconstruction from different viewpoints. These extensive experiment results **consistently suggest the superiority of our proposed method**, including the state-of-the-art perceptual performance and fairly remarkable generalization to challenging scenarios including in the presence of occlusion and less commonly seen shapes.
>
> Furthermore, we also conduct test-time optimization for baseline methods with access to silhouettes for a fair comparison. As the quantitative results are shown in Table 4 of the revised Supplementary Material, our proposed method remains highly competitive. It further demonstrates the effectiveness of incorporating generative prior in 3D reconstruction.
>
> **Q2: A fair comparison would be also test-time optimize CMR, UCMR, etc.**
>
> A2: Thank you for the suggestion. For a fair comparison, we have conducted experiments to test-time optimize CMR and UMR with access to masks, where the embedded feature extracted from the image encoder is updated towards minimizing the mask loss and texture loss. As shown in the table below, test-time optimization (TTO) of existing methods overall provides higher fidelity, but our method that exploits appealing generative prior remains highly competitive. Note that a relatively compact latent space is desirable for efficient optimization during the test time. Both CMR and UMR have a latent code with a dimension of 200. In contrast, U-CMR has a latent code with a dimension of 4096, whereas SMR does not follow an auto-encoder architecture, but directly encodes 3D attributes from the image with the associated mask. Therefore, SMR and U-CMR are infeasible to be adapted for test-time optimization. The experiment results and more details are also provided in Section 6 of the revised supplementary material.
>
> | Methods | IoU | FID_1 | FID_10 | FID_12 |
> |:-------------:|:-----:|:--------:|:-----:|:--------:|
> | CMR   | 0.703 | 140.9 | 176.2 | 180.1 |
> | CMR + TTO | 0.720 |  122.5 |  153.26 | 159.5 |
> | UMR  | 0.734 | 40.0 | 72.8 | 86.9 |
> | UMR + TTO | 0.742 | 38.7 | 78.9 | 90.2 |
> | ours | 0.708 | 38.6 | 38.6 | 56.6 |

---

> > ### Author Response · Authors · 2021-11-23
> > **Reply to Reviewer i1Mn [2/2]**
> >
> > **Q3: How would differentiable rasterization "introduce information loss" for Chamfer mask loss? How would both Chamfer losses deal with occlusion?**
> >
> > A3: The rasterization process discretizes the mesh into a grid of pixels. Therefore, the 2D mask distance may not accurately reflect the 3D ground-truth distance in 3D. In Section 5 of the revised Supplementary Material, we quantitatively analyze the accuracy of various mask losses by measuring the distance between two 3D shapes at different degrees of shape variations, where we show in Figure 2 that the quantization-induced loss error would be increasingly significant with smaller shape variation. This is particularly harmful to a well-trained ConvMesh, as can be seen from Table 3 of the Supplementary Material, a small perturbation in the latent space usually corresponds to a slight variation in the 3D shape, which might be detrimental for geometric learning.
> >
> > Occlusion is not a concern in both Chamfer losses. Chamfer mask loss intercepts differentiable rasterization, and is computed between the foreground pixels of the input mask and all the projected vertices including visible and invisible ones, as they all eventually fall within the rendered silhouette. Chamfer texture loss is computed after rasterization between all observed pixels or feature vectors, i.e. between the rendered image and the input image (foreground pixels only) for pixel-level loss, and feature maps extracted from these images for feature-level loss.
> >
> > **Q4: Are the weights of the generative network frozen during inference? Importance of the pre-trained network's accuracy. The extreme case of using a randomly initialized network for optimization.**
> >
> > A4: We keep the weights frozen when optimizing the latent code. The key of our approach is to incorporate the meaningful generative prior in a well-trained GAN. Hence the quality of a pre-trained network would largely affect the fidelity of 3D reconstruction. Unlike deep image prior [Ulyanov et al 2018] where an image can be reconstructed by tuning a randomly initialized CNN, lifting a 2D image into a 3D object is not trivial as there is additional information required.
> >
> > Ulyanov et al.: Deep image prior. In: CVPR 2018
> >
> > **Q5: The ablation result (Table 1) should normally go after the results (Table 2)**
> >
> > A5: Thanks for the suggestion, and we have placed the ablation study after other subsections in Section 4 of the revision.

---

> > > ### Comment · Reviewer_i1Mn · 2021-11-27
> > > **Response to the rebuttal**
> > >
> > > Thanks for providing a rebuttal.
> > > - The additional results of test-time optimizing the baselines partially address my concern on fairness of comparisons. Under the fair setup, the the proposed method achieves better FID scores than the baselines. But still, I'd like to see a visual comparison under this setup. Given that CMR also learns a shape prior from the same dataset as MeshInversion, I'm not convinced why CMR could not achieve decent quality by test-time optimizing the shape codes.
> > >
> > > Minor points
> > > - Since the paper is positioned as mesh reconstruction (not view synthesis), I still think a proper evaluation would include comparisons against 3D ground-truth. This can be done with synthetic data.
> > > - Agreed that the chamfer mask loss would not suffer from the quantization issue, but the difference would be rather minor, especially when the rendered image resolution is high. Also the fact that chamfer feature loss takes advantage of rasterization to determine the visible vertices is unsatisfactory (still acceptable), due to its non-differentiable nature.

---

### Official Review · Reviewer_LKvE · 2021-10-29

**Correctness:** 2
**Technical Novelty And Significance:** 2
**Empirical Novelty And Significance:** 2
**Recommendation:** 3
**Confidence:** 4

**Main Review:**

### Strengths
#### S1 - Novel Chamfer texture loss
- The novel Chamfer texture loss seems to be quite effective in maintaining high-fidelity textures in the presence of pixel misalignments. This can potentially be useful in many cases.

#### S2 - High quality results
- This test time optimization approach seems to produce better shape and texture reconstructions than existing single pass inference methods.
- By leveraging the prior captured during pretraining, the model is able to filter out out of distribution noise in the input image, eg. still producing a full textured bird in the presence of occlusion.

### Weaknesses
#### W1 - Core method similar to simple test-time optimization
- I do not see how the proposed MeshInversion is different from test time optimization. I think similar test time optimization can be easily done with existing methods to obtain higher fidelity results.
- The major claim is to leverage generative priors to achieve 3D GAN inversion. The only benefit of the generative prior that is evident here is perhaps the texture prior, ie., the fact that the model can produce full-textured bird under occlusion.
- However, the geometric prior is much more under-utilized. The model still requires camera/object poses and masks as input during inference.

#### W2 - Unfair comparisons, insufficient evaluation
- I do not find the comparisons convincing enough. The proposed method is based on test time optimization, whereas all compared methods perform inference with only a single forward pass. I think similar test time optimization could also be easily done with other methods, for example using BGFS search.
- 2D mask projection IoU is not indicative of the quality of 3D shape reconstruction. Especially, it is directly optimized as an objective during test time optimization.
- It is also unclear to me whether multi-view FID is really indicative of the 3D shapes, rather than being primarily sensitive to the texture. Table 1 shows the ablation models with a much lower mask IoU (0.57 vs. 0.71) still achieves relatively high multi-view FID score (55.7 vs. 45.4). I suspect the this metric might be more sensitive to texture than to the shape.
- Moreover, most of the reconstruction results, including the ones in the supplementary material, are visualized from the same viewpoint as the input image. Only few two-view examples are provided in Fig. 1 and Fig. 4. It is difficult to judge the quality of the 3D shapes. Extensive multi-view visualizations including untextured meshes should be provided, in order to judge the shape quality.
- The reconstructed meshes of cars in Fig. 6 also do not look very convincing. The paper claims that the proposed method predicts the mesh by deforming a sphere, whereas other methods requires template meshes for training, which I do not agree with. The proposed method requires pretraining the ConvMesh, which is essentially akin to a template mesh. And the method also assumes camera poses available, unlike U-CMR.
- The baseline with L1 and IoU mask losses seems to lead to poor reconstructions, indicated by the 0.57 mask IoU. This is surprising to me as it should be quite straightforward to minimize the mask loss during optimization. If the proposed Chamfer mask loss can lead to better fitting, I do not see why L1 or IoU loss cannot, with a reasonable weight and sufficient iterations.
- Regarding the user study, it is a great effort, but why not present the three meshes to the users, rather than only three views?

### Clarifications
- Are the camera poses fixed or jointly refined during inference?


**Summary Of The Paper:**

This paper presents a test-time optimization method that is able to produce a textured 3D mesh by fitting to a single input image, given a pretrained category-specific generative model of textured meshes and the estimated camera/object pose as input.

To enable high fidelity texture and shape reconstruction, the paper introduces a novel Chamfer texture loss that takes into account potential geometric misalignments, as well as a Chamfer mask loss that is said to encourage uniform deformation.

**Summary Of The Review:**

Overall, I do not find the major claim of exploiting 3D generative priors for single image mesh prediction clearly validated in this paper. The presented method still requires 2D mask and camera pose given as input, and ends up being similar to test-time optimization with existing methods. Moreover, the evaluation and comparisons are not entirely convincing. Therefore, I recommend reject.

---

> ### Author Response · Authors · 2021-11-23
> **Reply to Reviewer LKvE [1/3]**
>
> Thank you for your insightful comments. Below we address individual concerns. We also include our revision in the main paper and the supplementary material(content in purple).
>
> **Q1: I do not see how the proposed MeshInversion is different from test time optimization. I think similar test time optimization can be easily done with existing methods to obtain higher fidelity results.**
>
> A1: Thanks for raising this concern. The key insight is to incorporate the generative prior in a pre-trained GAN. To demonstrate its superiority over similar test time optimization methods, we also conduct test-time optimization (TTO) for baseline methods, if applicable. With access to the input image and mask, we fine-tune the embedded feature extracted from the image encoder to minimize the mask loss and texture loss. As shown in the table below, test-time optimization of baseline methods overall gives a higher fidelity, but our method remains highly competitive particularly for perceptual metrics. This fair comparison further shows the superiority of generative prior captured through adversarial training over that captured in the auto-encoder. The experiment results and more details are also provided in Section 6 of the revised supplementary material.
>
> | Methods | IoU | FID_1 | FID_10 | FID_12 |
> |:-------------:|:-----:|:--------:|:-----:|:--------:|
> | CMR   | 0.703 | 140.9 | 176.2 | 180.1 |
> | CMR + TTO | 0.720 |  122.5 |  153.26 | 159.5 |
> | UMR  | 0.734 | 40.0 | 72.8 | 86.9 |
> | UMR + TTO | 0.742 | 38.7 | 78.9 | 90.2 |
> | ours | 0.708 | 38.6 | 38.6 | 56.6 |
>
> **Q2: The only benefit of the generative prior that is evident here is perhaps the texture prior.**
>
> A2: As stated in the introduction, our proposed method gives SOTA appearance thanks to the meaningful texture prior, and gives geometric performance on-par with the existing CMR-based methods. Particularly, incorporation of generative prior gives reasonably good performance for less commonly seen shapes, such as birds with open wings (d of Figure 1, and d, e of Figure 3), where existing methods typically fail to generalize.
>
> **Q3: The model still requires camera/object poses and masks as input during inference.**
>
> A3: The involvement of a pre-trained GAN in our framework simplifies the task and allows us to train a camera pose estimator individually; the silhouette mask can be estimated by off-the-shelf instance segmentation methods. Since our focus in this work is to explore generative prior for 3D reconstruction, we base our experiments mainly on silhouette ground truths and cameras estimated by structure-from-motion (SfM), following the setting of SMR and DIB-R [Chen et al. 2019] respectively. In Section 4 of the revised Supplementary Material, we also validate the robustness of our method under relaxed conditions: with camera poses predicted by an off-the-shelf camera pose estimator from CMR, and with masks predicted by PointRend [Kirillov et al. 2020], which is pre-trained on COCO without fine-tuning. The results show that our method is **reasonably robust to inaccurately predicted camera poses and invariant to masks predicted by off-the-shelf instance segmentation methods**. The predicted cameras by CMR gives 6.03 degree of azimuth error and 4.33 degree of elevation error. The predicted masks by PointRend give IoU of 0.886.
>
> | Methods | IoU | FID_1 | FID_10 | FID_12 |
> |:-------------:|:-----:|:--------:|:-----:|:--------:|
> | Predicted cameras by CMR  | 0.703 | 43.1 | 44.1 | 59.9 |
> | Predicted masks by PointRend | 0.710 | 37.9 | 38.9 | 56.7 |
> | Cameras by SfM and ground-truth masks | 0.708 | 38.6 | 38.6 | 56.6 |
>
> Kirillov et al.: PointRend: Image segmentation as rendering. In: CVPR 2020
>
> Chen et al.: Learning to predict 3d objects with an interpolation-based differentiable renderer. In: NeurIPS 2019

---

> > ### Author Response · Authors · 2021-11-23
> > **Reply to Reviewer LKvE [2/3]**
> >
> > **Q4: Similar test time optimization could also be easily done with other methods, for example using BGFS search.**
> >
> > A4: Thanks for the suggestion. We have adapted test-time optimization (TTO) for baseline methods, if applicable, with access to masks. During inference, we fine-tune the embedded feature extracted from the image encoder towards minimizing the mask loss and texture loss. For an equal comparison, we fine-tune with the same Adam optimizer and for the same number of iterations, 200, for each testing instance. Since MeshInversion uses randomly initialized latent code whereas the forward pass by the image encoder already provides a good initialization, we use a smaller learning rate, 1e-4.
> >
> > As the results are shown below, test-time optimization overall gives a higher fidelity for baseline methods, but our method remains highly competitive especially for FIDs. Interestingly, UMR with test-time optimization achieves limited improvement in terms of IoU and single-view FID at the cost of worsening novel-view FID. This fair comparison further shows the superiority of generative prior captured through adversarial training over that captured in the auto-encoder. The experiment results and more details are also provided in Section 6 of the revised supplementary material.
> >
> > | Methods | IoU | FID_1 | FID_10 | FID_12 |
> > |:-------------:|:-----:|:--------:|:-----:|:--------:|
> > | CMR   | 0.703 | 140.9 | 176.2 | 180.1 |
> > | CMR + TTO | 0.720 |  122.5 |  153.26 | 159.5 |
> > | UMR  | 0.734 | 40.0 | 72.8 | 86.9 |
> > | UMR + TTO | 0.742 | 38.7 | 78.9 | 90.2 |
> > | ours | 0.708 | 38.6 | 38.6 | 56.6 |
> >
> > **Q5: 2D mask projection IoU is not indicative of the quality of 3D shape reconstruction. Especially, it is directly optimized as an objective during test time optimization.**
> >
> > A5: We agree with the reviewer, so improve the fairness of our evaluation from the following four aspects: 1) We quantitatively evaluate the multi-view FID to assess if the rendered images look realistic from different viewpoints. 2) We provide qualitative results from more novel views, with more results in Section 7 of the revised Supplementary Material. 3) We conduct a user study to evaluate various methods based on shape, appearance, and overall 3D reconstruction from different viewpoints. 4) Furthermore, we have also test-time optimized baseline methods with access to masks in Section 6 of the revised Supplementary Material. These extensive experiment results consistently show the relative advantage of our proposed method, including the state-of-the-art perceptual performance and fairly remarkable generalization to challenging scenarios including in the presence of occlusion and less commonly seen shapes.
> >
> > **Q6: Table 1 shows the ablation models with a much lower mask IoU (0.57 vs. 0.71) still achieves relatively high multi-view FID score (55.7 vs. 45.4). I suspect the this metric might be more sensitive to texture than to the shape.**
> >
> > A6: In fact, the relatively high multi-view FID is mainly a result of our Chamfer texture loss. Being robust to misalignment, it is still effective towards matching the appearance given mismatched shapes. We have also replaced our Chamfer texture loss with conventional texture losses (L1+ perceptual loss) which gives much worse FID scores. Note that we have updated the experiment results with Chamfer texture loss at both pixel and feature level (previously only pixel level). We have also included the results in Section 4 of the Supplementary Material.
> >
> > | Methods | Texture loss  | IoU | FID_1 | FID_10 | FID_12 |
> > |:-------------:|:-----:|:--------:|:-----:|:--------:|:--------:|
> > | IoU loss | L1 + perceptual loss | 0.588 | 62.3 | 60.6 | 76.3 |
> > | L1 loss | Chamfer texture loss (pixel- and feature- level) | 0.582  | 51.7 | 53.2 | 64.4 |
> > | IoU loss | Chamfer texture loss (pixel- and feature- level)| 0.605 | 51.2  | 50.8  | 62.0 |
> > | Chamfer mask loss | Chamfer texture loss (pixel- and feature- level) | 0.708 | 38.6 | 38.6 | 56.6 |
> >
> > **Q7: Extensive multi-view visualizations including untextured meshes should be provided.**
> >
> > A7: Thanks for the suggestion. More multi-view visualization results including untextured meshes for both birds and cars are included in Figure 4 and Figure 5 respectively, in the revised Supplementary Material.
> >
> > **Q8: The proposed method requires pretraining the ConvMesh, which is essentially akin to a template mesh.**
> >
> > A8: We agree with the reviewer. In fact, the appealing GAN prior implicitly provides a rich number of templates that makes it possible to reconstruct cars of different models. We have made it clearer in the revised Section 4.3.

---

> > > ### Author Response · Authors · 2021-11-23
> > > **Reply to Reviewer LKvE [3/3]**
> > >
> > > **Q9: Why L1 or IoU loss cannot lead to better fitting?**
> > >
> > > A9: Both L1 and Iou mask losses involve rasterization that discretizes the mesh into a grid of pixels, which inevitably introduces information loss and thus inaccurate supervision signals.
> > >
> > > In Section 5 of the revised Supplementary Material, we quantitatively analyze the accuracy of various mask losses by measuring the distance between two 3D shapes at different degrees of shape variations.
> > > Specifically, we utilize the pre-trained ConvMesh to generate random shapes and get the deviated shapes by introducing disturbance of varying size in the latent space. We take the 3D Chamfer distance between the original and deviated shapes as the "ground truth" distance, and plot these 2D distance-to-3D Chamfer distance ratios across different degrees of shape variations. Note that we take the L1 form for Chamfer distance and Chamfer mask loss. Therefore, all these L1-like 2D distances should ideally be linearly correlated to the 3D Chamfer distance.
> > >
> > > However, as shown in Figure 2 of the Supplementary Material, both IoU loss and L1 loss have a varying ratio to the ground truth distance at small shape variations, which implies that discretization-induced loss error is increasingly significant with smaller shape variation.
> > > This is particularly harmful to a well-trained ConvMesh, as can be seen from Table 3 of the Supplementary Material, a small perturbation in the latent space usually corresponds to a slight variation in the 3D shape, which might be detrimental for geometric learning in the course of GAN inversion.
> > >
> > > **Q10: Regarding the user study, it is a great effort, but why not present the three meshes to the users, rather than only three views?**
> > >
> > > A10: For the ease of distribution via a PDF file, we conduct the user study in the form of rendered images in three different views instead of original meshes.
> > >
> > > **Q11: Are the camera poses fixed or jointly refined during inference?**
> > >
> > > A11: We keep the camera poses fixed during inference, and we leave joint refinement to the future work.

---

> > ### Comment · Reviewer_LKvE · 2021-11-28
> > **Thanks for the response**
> >
> > I appreciate the authors' efforts in putting together these detailed responses.
> >
> > I still think there are a few major flaws in the evaluation of 3D shape reconstruction.
> > - There is not a convincing metric for evaluating the 3D reconstruction quality. I do not agree that multi-view FID is a fair metric for comparison. This metric is too sensitive to texture and does not properly evaluate shape reconstruction. All other methods, except for U-CMR predict texture flow and sample texture from the input image. The quality of the final texture thus depends not only on the shape, but also and perhaps more importantly on the pose and texture flow prediction. Unless all of them are perfectly accurate, the sampled texture would definitely appear broken. For example, a perfect shape reconstruction with a slightly off pose estimation could result in a poor score (high FID). Or due to the symmetry assumption enforced by all methods on the texture, the sampled texture of an articulated bird will certainly also look weird.
> > - Overall, all the qualitative and quantitative evidence points to better appearance modeling, but there is no clear evidence of better shape reconstruction than existing methods with test time optimization.
> >
> > The authors also stressed that this paper focused more on appearance modeling, and only claims on-par performance on the geometry. This claim would sound fine to me with the evidence demonstrated. Although, the method is termed "MeshInversion", which emphasizes "mesh" rather than appearance.
> >
> > However, more importantly, it becomes less interesting to me -- requiring (a) pre-computed results from an inverse-rendering model, (b) pre-training another GAN model, and (c) test-time optimization assuming mask and camera pose available, just to get improved texture. What would be much more exciting for me is to either (1) show much better shape reconstruction results as well, or (2) remove the need of mask and camera pose at test time by better exploiting the generative prior on geometry.
> >
> > I will raise my rating to 5 if the authors agree to carefully adjust the exposition to emphasize more on appearance modeling rather than geometry/mesh, but still think this falls short for an acceptance in terms of technical contribution.

---

### Official Review · Reviewer_owCV · 2021-11-02

**Correctness:** 2
**Technical Novelty And Significance:** 2
**Empirical Novelty And Significance:** 2
**Recommendation:** 5
**Confidence:** 4

**Main Review:**

### Strengths:
- GAN-inversion for 3D generative models is a very interesting research area to the community which is not yet well explored.
-  The qualitative results in the paper show impressive results on single-view 3D reconstruction.

### Weaknesses:
- The experiments do not sufficiently motivate the necessity of the proposed method:
    - Does the shape really need to be learned by inversion? ConvMesh already trains an encoder/decoder to estimate the shape from a single image – why not simply use this and optimize only for the texture? Did you try this and it did not work? Wouldn’t this greatly simplify the task? Are the proposed terms even needed in this scenario?
  - The baselines are not optimized specifically per evaluated image while the proposed method requires test-time optimization for every image. This difference should be made more clear. Also, is there a way to modify the baselines such that the comparison becomes more equal?
  - The quantitative comparison to the baselines shows mixed results. For the qualitative results, it is unclear how samples were selected. From this, it is hard to judge if the proposed method indeed outperforms existing methods.
- The limitations of this work are not discussed.
- The paper lacks structure and is not self-contained. Also, the motivation for the approach is not clear enough. See comments below.

### Missing related work
- Single-view 3D reconstruction:
using multi-view images: [1,2,3]
using radiance fields: [4]

> [1] Lior Yariv, Yoni Kasten, Dror Moran, Meirav Galun, Matan Atzmon, Ronen Basri, Yaron Lipman. Multiview Neural Surface Reconstruction by Disentangling Geometry and Appearance. NeurIPS 2020.
>
>[2] Peng Wang, Lingjie Liu, Yuan Liu, Christian Theobalt, Taku Komura, Wenping Wang. NeuS: Learning Neural Implicit Surfaces by Volume Rendering for Multi-view Reconstruction. arXiv preprint arXiv:2106.10689
>
>[3] Michael Oechsle, Songyou Peng, Andreas Geiger.UNISURF: Unifying Neural Implicit Surfaces and Radiance Fields for Multi-View Reconstruction. ICCV 2021
>
>[4] Konstantinos Rematas, Ricardo Martin-Brualla, Vittorio Ferrari. ShaRF: Shape-conditioned Radiance Fields from a Single View. ICML 2021

### Additional questions / comments:
- To me, the motivation is not quite clear. If I understand correctly, the main argument is that, in contrast to existing methods that rely on 2D data, this method can leverage a 3D prior. However, this prior is, similarly to existing works, learned from only 2D supervision, no? Should the key question then not rather be if a generative model provides a better prior than the auto-encoder framework which is used in most existing methods?
-  The main approach this work builds on [Pavllo et al.] is missing from the related work section entirely and it is not clear enough why a direct comparison to this method is not possible. I had to carefully read [Pavllo et al.] first to really understand this work, which ideally should not be the case for a self-contained paper.
- In the results in Figure 1 of the supplementary it looks like the prior does not always match the images well, particularly the shape e.g. in b), e), f) h) in which UMR appears to work better. Please discuss these limitations in the paper.
 - Baseline for Table 1: It would be interesting to combine perceptual + image based (L1/L2) as a potentially stronger baseline. This is particularly interesting since the perceptual loss already gets quite close to the performance of the proposed loss functions.
 - Please provide more qualitative results for Pascal 3D, particularly different viewpoints in the supplementary, as the ones in the paper are fairly close to the input views.
  - L_CM and L_CT are not formally defined in the paper, please provide explicit formulas for both loss terms as these are main contributions.
- It is unclear to me what ground truth from SfM means? If it is inferred with SfM it is not the ground truth of the image and if ground truth is available one does not need SfM?
- Could you explain how the camera pose estimator is trained ‘individually’ in your setting?
- Please add qualitative results for the ablation in Table 1 to the appendix rather than only describing it in the text.
- General: please add references to the appendix where appropriate, e.g. for the ablation on the discriminator in image space in sec 3.1.
- Sec 3.3 S_f is not defined

### Misc:
- Introduction, paragraph 4: “Chamfar Texture Loss”
- Related Work: ‘’It aims to recover”: informal
- Sec 3.4 “is shown”
- Sec 3.1 “ in the form of a pose-invariant”

**Summary Of The Paper:**

The work proposes inversion of a generative model for single-view 3D reconstruction.
Interestingly, the inversion is applied to a generative model that generates a 3D representation of an object rather than only a (2D) image.
To improve the inversion process from 2D to 3D, two Chamfer-distance inspired losses are proposed for texture and shape, respectively.

**Summary Of The Review:**

The problem setting is interesting and the qualitative results in the paper are impressive. However, the experiments appear not sound enough to support the necessity of the approach and important baselines seem to be missing. Hence I lean towards rejecting the paper.

---

> ### Author Response · Authors · 2021-11-23
> **Reply to Reviewer owCV [1/3]**
>
> Thank you for your comments. Below we address individual concerns. We also include our revision in the main paper and the Supplementary Material(content in purple). Training and evaluation code will be released to ensure reproducibility.
>
> **Q1: The necessity of the proposed method.**
>
> A1: As stressed in Paragraph 3 of Section 1, MeshInversion is an alternative approach to single-view 3D reconstruction. Instead of replacing existing methods, it provides complementary benefits in some aspects. The main value of this work is to reveal that 3D generative prior, even though learned from only 2D supervision, significantly benefits 3D reconstruction in various challenging scenarios including in the presence of complex textures, occlusion, and less commonly seen shapes. As for experiments supporting the proposed methods, we answer them separately in the following questions.
>
> **Q2: Why not simply use the encoder/decoder trained by ConvMesh for shape estimation and optimize only for the texture?**
>
> A2: Thank you for raising this concern. The encoder/decoder for ConvMesh serves the purpose of pseudo ground truth generation. Specifically, the pseudo UV space texture map and deformation map are derived from a form of inverse rendering of the training images, which requires a pre-trained shape encoder/decoder. With this purpose, it is overfitted to the training set, and does not generalize very well to unseen images. Quantitatively, the encoder/decoder gives IoU of 0.671 in contrast to MeshInversion with IoU of 0.708. We have made this clearer in revised Section 1 of the Supplementary Material. In addition, the generator of ConvMesh outputs both the UV texture map and UV deformation map from a shared backbone from a single latent vector. Therefore, even a good performing standalone shape encoder/decoder **would not simplify** the task, but it is infeasible to **fully disentangle** shape and texture for the generator.
>
> | Shape Estimation Approach | IoU |
> |:-------------:|:-----:|
> | encoder/decoder | 0.671 |
> | MeshInversion | 0.708 |
>
> **Q3: Is there a way to modify the baselines to test-time optimization such that the comparison becomes more equal?**
>
> A3: We thank the reviewer for the suggestion. We have adapted test-time optimization (TTO) for baseline methods, if applicable, with access to masks. During inference, the embedded feature extracted from the image encoder is updated towards minimizing the mask loss and texture loss. As the results are shown below, test-time optimization overall gives a higher fidelity for baseline methods, but our method remains competitive particularly in terms of FIDs. Interestingly, UMR with test-time optimization achieves limited improvement in terms of IoU and single-view FID at the cost of worsening novel-view FID. The experiment results and more details are also provided in Section 6 of the revised supplementary material.
>
> | Methods | IoU | FID_1 | FID_10 | FID_12 |
> |:-------------:|:-----:|:--------:|:-----:|:--------:|
> | CMR   | 0.703 | 140.9 | 176.2 | 180.1 |
> | CMR + TTO | 0.720 |  122.5 |  153.26 | 159.5 |
> | UMR  | 0.734 | 40.0 | 72.8 | 86.9 |
> | UMR + TTO | 0.742 | 38.7 | 78.9 | 90.2 |
> | ours | 0.708 | 38.6 | 38.6 | 56.6 |
>
> **Q4: The quantitative comparison to the baselines shows mixed results. For the qualitative results, it is unclear how samples were selected.**
>
> A4: The quantitative results are computed and reported in a comprehensive and consistent manner on the test split of CUB with metrics including IoU, FID_1, FID_10, and FID_12, where FID_12 is proposed by SMR, and FID_10 is proposed by us considering that the exact front view and back view or not common in CUB. The samples in the qualitative results were sampled to cover most representative scenarios, including complex and a wide range of texture, with highly articulated shapes, and in the presence of occlusion, etc. Training and evaluation code will be released to ensure reproducibility.
>
> **Q5: The limitations of this work.**
>
> A5: Currently, we adopt a test-time optimization-based approach to exploit the generative prior, which may be costly for some real applications.
>
> **Q6: Missing related work.**
>
> A6: Thanks for these papers, which are included in the revised related work. We note that [1,2,3] require multiview images and [4] requires 3D ground truths supervision during training, whereas we address a more challenging task where each instance has only a single-view observation in the training set.

---

> > ### Author Response · Authors · 2021-11-23
> > **Reply to Reviewer owCV [2/3]**
> >
> > **Q7: To me, the motivation is not quite clear.**
> >
> > A7: The key motivation in this work is that exploiting generative prior encapsulated in a pre-trained GAN would benefit 3D reconstruction. Although the 3D GAN is trained with 2D data as well, it gives superior prior compared to the auto-encoder framework, even though with additional test-time optimization. Moreover, while using such a pre-trained 3D GAN already leads to good results, replacing it with a 3D GAN pre-trained directly on 3D data, if available, is expected to further boost the performance without changing our method.
> >
> > **Q8: [Pavllo et al.] is missing from the related work section entirely and it is not clear enough why a direct comparison to this method is not possible?**
> >
> > A8: Thanks for raising this concern, and we have updated Section 1 of the Supplementary Material to make it self-contained. We would like to draw the reviewer’s attention that ConvMesh [Pavllo et al.] is a generative model for free generation of 3D objects, and it cannot be directly used for 3D reconstruction given a single-view image, whereas a feasible approach to incorporating a pre-trained GAN for reconstruction is through GAN inversion, ie, finding the latent code (noise vector) that corresponds to the target 3D object, and this is exactly how we realize 3D reconstruction with pre-trained ConvMesh. Moreover, our method is not limited to ConvMesh. We use it because it is currently a representative pre-trained GAN.
> >
> > **Q9: In Figure 1 of the supplementary, it looks like the prior does not always match the images well, particularly the shape e.g. in b), e), f) h) in which UMR appears to work better.**
> >
> > A9: Our method is an alternative rather than a replacement to existing baselines. One of the major relative advantages is that it generalizes reasonably well to less commonly seen shapes, such as birds with extended tails, e.g. c) of Figure 1 of the main paper, and open wings, e.g. d) of Figure 1, d) and e) of Figure 3 of the main paper. Another major relative advantage is that it gives realistic recovery of invisible regions even in the presence of occlusion, such as b) and f) of Figure 2 of the supplementary (previous Figure 1), where texture flow-based methods, i.e. UMR, CMR and SMR, typically fail to do so.
> >
> > **Q10: It would be interesting to combine perceptual + image based (L1/L2) as a potentially stronger baseline.**
> >
> > A10: Thanks for the suggestion. As shown in the table below, combining L1+ perceptual loss does provide a stronger baseline.
> > For a fair comparison, we implement our Chamfer texture loss at both the pixel and the feature level (previously only pixel level). This not only demonstrates that our formulation of the Chamfer texture loss is readily extendable to feature maps, but also leads to a new SOTA for the perceptual metrics. We have updated Section 3.2 and Section 3.3 for the losses, and updated the results in Table 3 (previous Table 1).
> >
> > | Texture loss | Mask loss | IoU | FID_1 | FID_10 | FID_12 |
> > |:-------------------------:|:-----:|:-----:|:--------:|:-----:|:--------:|
> > | L1  loss | Chamfer mask loss | 0.705 | 71.2 | 97.3 | 108.4  |
> > | perceptual loss | Chamfer mask loss | 0.701 | 49.8  | 52.3  | 69.5 |
> > | L1 + perceptual loss | Chamfer mask loss | 0.707  | 47.1  | 50.8 | 66.7 |
> > | Chamfer texture loss pixel | Chamfer mask loss | 0.708 | 47.9  | 45.4  | 63.7 |
> > | Chamfer texture loss pixel + feature | Chamfer mask loss | 0.708 | 38.6 | 38.6 | 56.6 |
> >
> > **Q11: More qualitative results for Pascal 3D.**
> >
> > A11: More novel-view results for Pascal 3D cars are included in Figure 5 of the revised Supplementary Material.
> >
> > **Q12: L_CM and L_CT are not formally defined in the paper.**
> >
> > A12: Chamfer Mask Loss L_CM and Chamfer Texture Loss L_CT are defined in Section 3.3 and Section 3.2 respectively.
> >
> > **Q13: It is unclear to me what ground truth from SfM means?**
> >
> > A13: Thanks for pointing this out. There are no absolute ground truth cameras. The camera annotations are roughly estimated by applying SfM to the annotated keypoint locations, which can be found in the CMR paper. We have revised the experimental setting.
> >
> > **Q14: How the camera pose estimator is trained ‘individually’**
> >
> > A14: As stated in the Experimental Setting, camera pose estimation is not the focus of this study and we base our experiments mainly on cameras estimated by SfM. In Section 4 of the Supplementary Material, we have also validated the robustness of our framework under camera poses predicted by an off-the-shelf camera pose estimator from CMR, which gives 6.03 degrees of azimuth error and 4.33 degrees of elevation error, compared to those estimated by SfM. We have also made this clearer in the revised Experimental Setting section.

---

> > > ### Author Response · Authors · 2021-11-23
> > > **Reply to Reviewer owCV [3/3]**
> > >
> > > **Q15: Add references to the appendix for the ablation on the discriminator in image space in sec 3.1**
> > >
> > > A15: Thanks for the suggestion, and we have updated it in Section 3.1.
> > >
> > > **Q16: Sec 3.3 S_f is not defined.**
> > >
> > > A16: S_f is the set of foreground pixels' normalized coordinates of the mask. It is defined in the revised Section 3.3.

---

> > > > ### Comment · Reviewer_owCV · 2021-11-30
> > > > **Response to Rebuttal**
> > > >
> > > > Thank you for addressing my concerns in detail.
> > > > However, I think particularly when comparing to the baselines which were optimized at test time the performance of this is approach is not strong enough wrt 3D reconstruction. Further, I share the concerns of the other reviewers regarding evaluation metrics for 3D reconstruction. Therefore, I decided to stick to my initial rating.

---

### Official Review · Reviewer_pz7V · 2021-11-03

**Correctness:** 4
**Technical Novelty And Significance:** 3
**Empirical Novelty And Significance:** 3
**Recommendation:** 6
**Confidence:** 4

**Main Review:**

- The paper proposes a very interesting alternative to the popular CMR formulation, which achieves good results, that look very realistic, both in terms of the texture and in terms of the shape of the objects (particularly birds). Quantitative results are also strong. Results on perceptual studies are also included.

- The setting of GAN inversion, to the best of my knowledge, is novel for the CMR setting. I find the idea interesting and I think it is well executed here.

- The code will be released, which further helps comparisons with future works.

- Most CMR methods are regression-based at test time, without any test-time optimization. It would be helpful to clarify this again at the experimental section, because this is a significant shift in the methodology. For example the mask is used as input at test time, which happens for SMR, but not CMR. I think the authors should highlight these differences more clearly.

- I would like some clarifications on the setting of the ablation study of Table 1. For the rows with Texture Losses, do the authors use their Chamfer Mask Loss (and vice versa, for the rows corresponding to the Mask Losses, do they also use their Texture Mask Loss)? Similarly, the last line is using the "predicted camera poses" or not?

- I would encourage the authors to provide more results from novel views. Currently, most results are from the side view of the bird. It would be useful to see more visualizations from other views as well (top/frontal).

**Summary Of The Paper:**

This paper addresses the problem of learning single view reconstruction for a specific category using an image collection. This is related to the CMR framework by Kanazawa et al, ECCV 2018. The main contribution of this paper is using a GAN, and at test time searching for the latent code that best resembles the object in the input image. This is done with test-time optimization, and two losses, a Chamfer Texture Loss and a Chamfer Mask Loss, are proposed for this.

**Summary Of The Review:**

Overall, I am positive about the work and I find the idea novel in the CMR setting. I am rating this paper with a 6, since I would like the authors to clarify some experimental details, but all in all, I expect to keep my positive score.

---

> ### Author Response · Authors · 2021-11-23
> **Reply to Reviewer pz7V**
>
> Thank you for your constructive comments. Below we address individual concerns. We also include our revision in the main paper and the supplementary material(content in purple).
>
> **Q1: Most CMR methods are regression-based at test time, without any test-time optimization. I think the authors should highlight these differences more clearly.**
>
> A1: Thank you for the suggestion, and we have made this clearer in the updated Figure 1 and Section 4.1. Moreover, we have also adapted test-time optimization (TTO) for baseline methods with access to the mask and input image, for a fair comparison. During inference, we fine-tune the embedded feature extracted from the image encoder towards minimizing the mask loss and texture loss. As shown in the table below, TTO overall gives higher fidelity for baseline methods, but our proposed method remains highly competitive especially in terms of perceptual metrics. Interestingly, UMR with test-time optimization achieves limited improvement in terms of IoU and single-view FID at the cost of worsening novel-view FID. This further shows the superiority of generative prior captured through adversarial training over that captured in the auto-encoder, and its effectiveness of such appealing prior in 3D reconstruction.
>
> | Methods | IoU | FID_1 | FID_10 | FID_12 |
> |:-------------:|:-----:|:--------:|:-----:|:--------:|
> | CMR   | 0.703 | 140.9 | 176.2 | 180.1 |
> | CMR + TTO | 0.720 |  122.5 |  153.26 | 159.5 |
> | UMR  | 0.734 | 40.0 | 72.8 | 86.9 |
> | UMR + TTO | 0.742 | 38.7 | 78.9 | 90.2 |
> | ours | 0.708 | 38.6 | 38.6 | 56.6 |
>
> **Q2: Clarifications on the setting of the ablation study of Table 1.**
>
> A2: When evaluating various texture losses, we use our proposed Chamfer mask loss, and vice verse, to better assess the effectiveness of our proposed Chamfer-based losses. It is worth noting that the formulation of our Chamfer texture loss is readily extendable to feature maps. Applying both pixel-level and feature-level Chamfer texture losses leads to a new SOTA for the perceptual metrics. We have made it clearer in the revised Table 3 (previous Table 1), and we have also reported more comprehensive ablation studies in Table 2 of the revised Supplementary Material, including a version of our method without neither the Chamfer mask loss nor the Chamfer texture loss.
>
> Since the main focus of this work is to study the effectiveness of exploiting GAN prior in 3D reconstruction, we base our experiments mainly on silhouette ground truths and cameras **estimated** by structure-from-motion (SfM), following the setting of SMR and DIB-R [Chen et al. 2019] respectively. As the results are shown below, we have also validated that our method is **reasonably robust to predicted camera poses** predicted by an off-the-shelf camera pose estimator from CMR, and is invariant under masks predicted by instance segmentation method PointRend [Kirillov et al. 2020] pre-trained on COCO without fine-tuning. The results are also included in Section 4 of the Supplementary Material.
>
> | Methods | IoU | FID_1 | FID_10 | FID_12 |
> |:-------------:|:-----:|:--------:|:-----:|:--------:|
> | Predicted cameras by CMR  | 0.703 | 43.1 | 44.1 | 59.9 |
> | Predicted masks by PointRend | 0.710 | 37.9 | 38.9 | 56.7 |
> | Cameras by SfM and ground-truth masks | 0.708 | 38.6 | 38.6 | 56.6 |
>
> Kirillov et al.: PointRend: Image segmentation as rendering. In: CVPR 2020
>
> Chen et al.: Learning to predict 3d objects with an interpolation-based differentiable renderer. In: NeurIPS 2019
>
> **Q3: More results from novel views.**
>
> More novel-view results for both birds and cars are included in Figure 4 and Figure 5 respectively, in the revised Supplementary Material.

---

> > ### Comment · Reviewer_pz7V · 2021-11-29
> > **Response to the rebuttal**
> >
> > I thank the authors for their response. They have considered most of my concerns and questions. I think that the discussion about the use or not of test-time optimization is very helpful, and I am satisfied that this is addressed more explicitly in the text and the extra experiment. I acknowledge also the comments from the other reviewers, and based on the results, it indeed looks like appearance is improving a lot, but 3D shape recovery is not benefiting that much. However, I still think that there is merit in this paper, and this is a promising direction, so I will keep my Weak Accept rating.

---

### Decision · Program_Chairs · 2022-01-20

**Decision:**

Reject

**Comment:**

This submission received 4 diverging ratings: 3, 5, 5, 6. On the positive side, reviewers appreciated the novelty of the approach and strong empirical performance. At the same time, all negatively-inclined reviewers mentioned unfair comparisons with baselines (which was partially addressed in the rebuttal), flaws in the evaluation protocols and ablations not fully supporting claims made in the paper. After discussions with the authors most reviewers decided to stick with their original ratings.
AC agrees that the remaining open questions around empirical validation will need to be answered more clearly before the paper can be accepted. The final recommendation is reject.